# Triacylglycerol and Fatty Acid Compositions of Blackberry, Red Raspberry, Black Raspberry, Blueberry and Cranberry Seed Oils by Ultra-Performance Convergence Chromatography-Quadrupole Time-of-Flight Mass Spectrometry

**DOI:** 10.3390/foods10112530

**Published:** 2021-10-21

**Authors:** Yinghua Luo, Fanghao Yuan, Yanfang Li, Junyi (Danny) Wang, Boyan Gao, Liangli (Lucy) Yu

**Affiliations:** 1College of Food Science and Nutritional Engineering, National Engineering Research Center for Fruit and Vegetable Processing, Key Laboratory of Fruit and Vegetable Processing Ministry of Agriculture, Engineering Research Centre for Fruit and Vegetable Processing, Ministry of Education, China Agricultural University, Beijing 100083, China; luoyinghua@cau.edu.cn; 2Institute of Food and Nutraceutical Science, School of Agriculture & Biology, Shanghai Jiao Tong University, Shanghai 200240, China; yuanfanghao@sjtu.edu.cn (F.Y.); zoe_li@sjtu.edu.cn (Y.L.); 3Bluewood Associaes Co., Ltd., Suzhou 215134, China; danny.wang@bwdasso.com; 4Department of Nutrition and Food Science, University of Maryland, College Park, MD 20742, USA; lyu5@umd.edu

**Keywords:** triacylglycerol (TAG) compositions, berry seed oils, ultra-performance convergence chromatography (UPC^2^), quadrupole time-of-flight mass spectrometry (QTOF MS)

## Abstract

The triacylglycerol (TAG) compositions of blackberry, red raspberry, black raspberry, blueberry and cranberry seed oils were examined using ultra-performance convergence chromatography-quadrupole time-of-flight mass spectrometry (UPC^2^-QTOF MS). A total of 52, 53, 52, 59 and 58 TAGs were detected and tentatively identified from the blackberry, red raspberry, black raspberry, blueberry and cranberry seed oils, respectively, according to their accurate molecular weight in MS1 and fragment ion profiles in MS2. OLL was the most abundant TAG in the blackberry, red raspberry and black raspberry seed oils. Furthermore, the fatty acid compositions of the five berry seed oils were directly determined by gas chromatography coupled with mass spectrometry (GC-MS). In addition, the seed oils had total phenolic contents ranging 13.68–177.06 µmol GAE (gallic acid equivalent)/L oil, and significant scavenging capacities against DPPH, peroxyl, and ABTS^+^ radicals. These results indicated that the combination of UPC^2^ and QTOF MS could effectively identify and semi-quantify the TAGs compositions of the berry seed oils with sn-position information for the fatty acids. Understanding the TAGs compositions of these berry seed oils could improve the utilization of these potentially high nutritional value oils for human health.

## 1. Introduction

Berries, including cranberry (*Vaccinium macrocarpon*), raspberry (*Rubus idaeus*), blackberry (*Rubus fruticosus*), blueberry (*Vaccinium corymbosum*) and strawberry (*Fragaria* × *ananassa*), are a group of widely consumed fruits globally. Beside the sweet and sour tastes, berry fruits are well welcomed for particular nutraceutical values in delaying the aging process and reducing the risk of several human chronic diseases [1,2,3]. To date, these berry fruits are known for their potential antioxidant [4,5,6,7,8], anti-inflammatory [9,10] and antimicrobial properties [5,11,12,13]. These health benefits enhanced the increasing market demand of berry fruits, but the juicy ovary of berry fruits make them difficult to be transported over long distances. In order to extend the application and increase the shelf life of berry fruits, processed berry products have become more and more popular in recent years. Major berry products, including berry wines, beverages, jams and smoothies, result in large amount of their processing byproducts, such as the skin and seeds. The utilizations of the byproducts not only could increase the overall commercial value of berry fruits, but also potentially develop novel food products with high nutritional value.

Berry seeds are known to be rich in minor nutrients and other health beneficial components [14]. These berry seeds are recognized as possible good sources of health-promoting oils due to their great contents of n-3 and total polyunsaturated fatty acids (PUFAs), tocopherols and phytosterols [15,16,17]. In one study, van Hoed et al. investigated the fatty acid compositions of blackberry, red raspberry, cranberry, blueberry and strawberry seed oils, and reported that the polyunsaturated fatty acids (PUFA) contents were more than 65% among the overall fatty acids in all the five berry seed oils, with 17.60–36.48% of α-linolenic acid, an n-3 essential fatty acid [18]. In 2014, Radočaj et al. reported PUFA contents of about 76.4% and 86.2% for blackberry and raspberry seed oils, respectively [19]. A few other studies also detected high n-3 and total PUFA contents in berry seed oils [16,17,18,19,20,21]. However, all these previous studies about berry seed oil compositions were focused on their fatty acid composition. None of those previous studies investigated the TAG compositions with sn-location information for the n-3 and other fatty acids. It is known that the sn-position of fatty acids in the TAGs plays a very important role in determining their absorption and bioavailability, and consequently their nutraceutical values [22,23,24,25]. For example, the sn-position of palmitic acid in triacylglycerol of milk could significantly change the lipids and calcium absorption, thus affecting fecal conditions of infants [23,24]. Therefore, understanding the sn-positions of the fatty acids is important for their functionality and behavior of the TAG in food systems.

Therefore, the present study was conducted to investigate the TAG compositions of blackberry, red raspberry, black raspberry, blueberry and cranberry seed oils using a supercritical CO_2_ ultra-performance convergence chromatography (UPC^2^) coupled with high resolution quadrupole time-of-flight mass spectrometry (Q TOF MS). The fatty acid compositions of the five berry seed oils were also determined with a gas chromatography coupled with mass spectrometry (GC-MS) to verify the UPC^2^-Q TOF MS results. In addition, the total phenolic contents, as well as the radical scavenging activities of the seed oils, were examined. The results could be used to promote the future nutraceutical development and/or functional food investigation from the seed oils.

## 2. Materials and Methods

### 2.1. Materials and Reagents

Blackberry, red raspberry, black raspberry, blueberry and cranberry seed oil samples were gifted from the Botanic Innovations (Spooner, WI, USA) and stored at −20 °C before analyses. All the LC mobile phases and essential chemical agents, including acetonitrile, methanol, isopropanol, and ammonium formate were purchased from Sigma-Aldrich (St. Louis, MO, USA) with a liquid chromatography coupled with mass spectrometry (LC-MS) purity. CO_2_ was food-grade with purity over 99.99% and obtained from Zhenxin Gas Co. Ltd. (Shanghai, China). Ultrapure water was purified with a Millipore Milli-Q 10 system (Billerica, MS, USA) in the laboratory. Fatty acid methyl ester (FAME) mixed standard (containing 37 FAMEs) was purchased from NU-CHEK Prep, Inc. (Elysian, MN, USA).

### 2.2. Sample Preparations

All the five seed oil samples were prepared based on our previous published methods [26,27,28,29]. One aliquot of 10 µL of each berry seed oil sample was mixed with 990 µL of acetonitrile/methanol/isopropanol (10:9:1, *v*/*v*/*v*), vortex-mixed for 20 s and centrifuged at 2000 rpm for 5 min at an ambient temperature. After removing supernatant, the residue was dissolved in 990 µL of isopropanol and injected into UPC^2^-Q TOF MS for analysis; each sample was prepared in triplicates. 

### 2.3. UPC^2^-MS System Condition

The conditions of UPC^2^-Q TOF MS were set according to a previously reported laboratory procedure [26,27,28,29]. Waters Acquity UPC^2^ system (Milford, MA, USA) equipped with an Acquity UPC^2^ BEH HSS C18 column (150 mm × 3.0 mm i.d.; 1.7 µm) was utilized for the separation of TAGs. All the other instrument conditions were similar as previous reported method. A Waters 1525 single pump was used as the compensated pump, and pumped 0.3 mL/min of 0.1% ammonium formate in methanol into the MS source. The TAG compositions of the berry seed oils were analyzed and semi-quantified with a Waters Xevo-G2 Q-TOF MS system as previously described. The ion mode, capillary voltage, cone voltage, temperatures of source and desolvation, as well as other parameters were similar as our previous publications. The collision energy was 6 eV in MS1, and information of fragment ions was collected in the MS2 mode and the collision energy was 35 eV, with the scan time set at 0.2/s.

### 2.4. Fatty Acid Composition Analysis

Fatty acid compositions of blackberry, red raspberry, black raspberry, blueberry and cranberry seed oils were determined according to a previously reported laboratory protocol [26,27,28,29]. One aliquot of 20 mg of each berry seed oil sample was mixed with 0.4 mL of methylbenzene and 0.4 mL of KOH-MeOH (0.5 mol/L). The mixture was heated at 60 °C for 10 min and cooled down to ambient temperature. After that, 2 mL of boron trifluoride-MeOH (14%) was added into the mixture that was kept at 60 °C for 5 min. Then, 2 mL of isooctane and 3 mL of ultrapure water were added and vortexed to end the reaction. The supernatant was injected for GC analysis after removing the moisture. The Agilent 7890A gas chromatograph with flame ionization detector (FID) detector and DB-23 silica capillary column (60 m length × 0.25 mm with a 0.25 µm film thickness) was used. Fatty acid methyl esters were identified by comparing the retention time of each peak in the berry seed oil samples with that of the FAME standards. And area normalization method was used to calculate the relative concentrations of fatty acids. 

### 2.5. Total Phenolic Content (TPC) and Radical Scavenging Activity Assays

The TPC and radical scavenging activity of five berry seed oils were also measured to clarify the potential bioactivity of these oils. In this study, three different radical scavenging assays were utilized to evaluate the radical scavenging activities of berry seed oils, including ABTS radical cation scavenging capacity, relative DPPH radical scavenging capacity (RDSC) and oxygen radical absorbance capacity (ORAC). All the assays were performed following the laboratory protocols previously published [30,31]. In general, 500 µL of each berry seed oil was mixed with 500 µL of 50% acetone. After vortexing for 20 s, the mixture was centrifuged at 8000 rpm for 5 min at an ambient temperature. The 50% acetone layer was collected, and the oil layer was re-extracted with the same procedure two more times. The three 50% acetone extractions were combined for further analyses. Each oil sample was extracted in triplicate.

### 2.6. Statistical Analysis

The relative concentrations of TAGs, fatty acids, total phenolic contents and antioxidant activities were reported as the mean ± standard deviation (SD). One-way ANOVA and Tukey’s post hoc test were utilized in SPSS 18.0 (Chicago, IL, USA) to analyze both triacylglycerol and fatty acid compositions, with *p* < 0.05 considered a significant difference.

## 3. Results and Discussion

### 3.1. Identification of TAGs from the Five Berry Seed Oils

All the TAGs detected were identified as quasi-molecular ions of [M + NH_4_]^+^ in the Q TOF MS positive ion mode. A total of 52, 53, 52, 59 and 58 TAGs were detected and tentatively identified from the blackberry, red raspberry, black raspberry, blueberry and cranberry seed oils, respectively (Figure 1). The chemical structures of TAGs were determined by analyzing their high resolution molecular weight and the mass fragment ion information. Every TAG contains three fatty acids, the initial letter was used to represent a fatty acid, and XYZ represents the structure of a TAG with X, Y and Z fatty acids in its sn-1,2,3 positions. For instance, POL stands for the structure of 1/3-palmitoyl-2-linoleicoyl-1/3-oleoylglycerol. Generally, there are three types of TAGs based on the possible fatty acid profiles: three of the same fatty acids such as LLL, two same and one different fatty acids such as OLL, and three different fatty acids such as POL. The chemical structures of all the TAGs were characterized with the elucidation of following three typical TAGs.

The peak at the retention time 6.94 min is a representative TAG with similar fatty acids in all the three sn-positions. The quasi-molecular ion of this TAG is [M + NH_4_]^+^ at *m*/*z* 902.8221 (Figure 2), the molecular formula could be calculated as C_57_H_104_O_6_ based on MS1 information. The MS fragments in its MS2 showed that there was only one fragment ion peak [M-RCOO + H]^+^ at *m*/*z* 603.5405, which corresponding to a diacyl fragment OO^+^ resulted from the TAG eliminating a oleic acid (with the *m*/*z* 299.2816). Taken together, this peak could be identified as an OOO TAG.

The peak at the retention time 6.58 min is an example for identifying TAG with two different fatty acids. The quasi-molecular ion of this TAG is [M + NH_4_]^+^ at *m*/*z* 900.8047 (Figure 3), the molecular formula was calculated as C_57_H_102_O_6_ from the MS1 information. The MS fragment from MS2 showed two major fragment ion peaks [M-RCOO + H]^+^ at *m*/*z* 601.5235 and 603.5349, which correspond to diacyl fragments LO^+^ and OO^+^ from the TAG by eliminating an oleic acid (with the *m*/*z* 299.2816) and a linoleic acid (with the *m*/*z* 297.2698), respectively. Due to that the relative natural abundances of LO^+^ and OO^+^ are 100% and 43% (Figure 3), and the well accepted knowledge that fatty acid on sn-2 position might have higher bond energy and is harder to be eliminated [29], this TAG is tentatively identified as LOO. 

The peak with a retention time of 6.30 min is an example for identification of TAG with three different fatty acids. The quasi-molecular ion of this TAG is [M + NH_4_]^+^ at *m*/*z* 874.7861 (Figure 4), and the molecular formula could be calculated as C_55_H_100_O_6_ based on the MS1 information. MS fragments in the MS2 showed three fragment ion peaks [M-RCOO + H]^+^ at *m*/*z* 601.5202, 577.5190 and 575.5037, respectively. Therefore, the eliminated fatty acid fragments could be calculated as *m*/*z* 273.2659, 297.2671 and 299.2824, which could be identified as palmitic, linoleic and oleic acids, respectively. The three diacyl fragments was identified as OL^+^, PO^+^ and PL^+^, and their natural abundances were 100%, 69% and 46%, respectively. The MS fragment PL^+^ had the lowest natural abundance, suggesting that oleic acid has the highest bond energy and is the most difficult to be eliminated; this TAG was tentatively identified as POL or LOP.

According to the same rule, the TAG peaks in the five berry seed oils were tentatively identified. Their peak areas were also measured, and the concentrations of each TAG in a berry seed oil were semi-quantified using an area normalization method. The types of TAGs and their relative concentrations in every berry seed oil are discussed separately in the following paragraphs.

There were, in total, 52 TAGs identified from blackberry seed oil. The three primary TAGs in the blackberry seed oil were OLL, LLLn and LnLnL (Table 1), with the relative concentrations at 12.31, 8.66 and 8.51 g/100 g total TAGs, respectively. Interestingly, similar trends were observed in the TAGs compositions of the red raspberry and black raspberry seed oils. The primary three TAGs in these two raspberry seed oils were also OLL, LLLn and LnLnL, the same as that in the blackberry seed oil. The concentrations of these three TAGs were 14.97, 9.67, and 9.04 g/100 g total TAGs in red raspberry seed oil, and 11.89, 11.67 and 9.95 g/100 g total TAGs in black raspberry seed oil. These three berry seed oils were similar in their TAG compositions, especially for the TAGs with a concentration more than 5 g/100 g total TAGs. This might be explained by the fact that the three berries belong to a same *Rosaceae* family *Rubus* genus, and with a similar biological botanical relationship.

A total of 59 TAGs were identified from the blueberry seed oil. Among all the TAGs, LLLn, LOO and LnLnL were the three primary ones, with relative concentrations at 9.71, 9.54 and 8.41 g/100 g total TAGs, respectively. For cranberry seed oil, 58 TAGs were separated and identified, and LOO (11.64 g/100 g), LOLn (10.65 g/100 g) and LnLnL (8.41 g/100 g) were the three greatest TAGs.

Since every TAG has three bounded fatty acids, the compositions of all the TAGs could be used to calculate the fatty acid compositions in berry seed oils. Based on the present TAG composition results, linoleic acid and linolenic acid are the two primary fatty acids in all the five berry seed oils. Linoleic acid concentrations were more than 50 g/100 g total TAGs in blackberry, red raspberry and black raspberry seed oils, and more than 30 g/100 g total TAGs in the blueberry and cranberry seed oils. It is important that α-linolenic acid levels were more than 30 g/100 g total TAGs in all the five berry seed oils, suggesting that these berry seed oils may serve as excellent dietary sources for this n-3 essential fatty acid and may be included in human diet for improving the n-6/n-3 ratio. These phenomena showed that all these berry seed oils contained great amounts of multi-unsaturated fatty acids (MUFA), which have potential abilities to be developed for food supplements or nutrients. 

### 3.2. Identifications of Fatty Acids from Five Berry Seed Oils

The fatty acid compositions of five berry seed oils were also directly examined using GC-MS, and five major fatty acids, including palmitic (16:0), stearic (18:0), oleic (18:1), linoleic (18:2) and linolenic (18:3) acids were identified and semi-quantified (Table 2). All the five berry seed oils contained over 90% of unsaturated fatty acids on a per weight basis, and less than 10% of saturated fatty acids. Among the saturated fatty acids, palmitic acid content was 2–3 times greater than that of stearic acid in all the five berry seed oils. Linoleic acid content was the greatest among all unsaturated fatty acids and accounted for over 50% of the total fatty acids in the blackberry, red raspberry and blackberry seed oils, in excellent agreement to the fatty acid profiles calculated from the TAG compositions (Table 2). These fatty acid compositions also agreed to that of the blackberry and raspberry seed oils reported previously [18,19,20,32]. In the present study, α-linolenic acid had the greatest concentration and consisted of about 50% of the total fatty acids in the blueberry and cranberry seed oils, followed by linoleic acid at a level of about 40 g/100 g total fatty acids. The fatty acid results obtained by GC-MS analysis supported the fatty acid compositions calculated from the TAGs (Table 2). It is understandable that minor amounts of fatty acids identified in the TAGs cannot be detected by GC-MS due to the detection limits of the two methods, since the limit of detection of fatty acids in the GC method are generally at microgram per milliliter, which is much higher than that in UPC^2^-MS. The sample preparation might also contribute to the different fatty acid profile data by GC-MS and UPC^2^-MS.

### 3.3. Total Phenolic Contents and Radical Scavenging Activities of Five Berry Seed Oils

The total phenolic contents and radical scavenging activities of the five berry seed oils were investigated in this study (Figure 5). The seed oils differed in their TPC values, ranging from 13.68 to 177.06 µmol GAE (gallic acid equivalent)/L oil. Red raspberry seed oil showed the greatest TPC value, followed by black raspberry seed oil at 68.62 µmol GAE/L oil, blackberry seed oil at 52.72 µmol GAE/L oil, blueberry seed oil at 24.16 µmol GAE/L oil and cranberry seed oil. The red raspberry seed oil was the most rich in phenolic compounds under the experimental conditions.

Different radical scavenging activity assays could be used to evaluate the radical scavenging abilities of a sample against different types of free radicals. In general, two or more assays are needed to draw a common conclusion for a radical scavenging sample because of the possible interference of the assay system. In this study, three different assays were applied to determine the radical scavenging activities of berry seed oils. Generally, all the tested berry seed oil samples showed the greatest abilities in competitively inhibiting oxygen radicals, since the ORAC values for all the berry seed oils were much greater than the results of other two assays. The blackberry seed oil showed the greatest radical scavenging activity, followed by red raspberry in all three assays. The blueberry and blackberry seed oils showed greater ABTS^•+^ and DPPH^•^ scavenging activities than the cranberry seed oil, but their ORAC values were smaller than that of the cranberry seed oil. These results indicated that different berry seed oils might contain different antioxidant capabilities and differed in their interactions with individual radical systems.

## 4. Conclusions

In summary, this study examined the TAG compositions and the sn-positions of individual fatty acids of the five berry seed oils for the first time. Linoleic (18:2n-6) and α-linolenic (18:3n-3) acids, the two essential PUFAs, accounted for more than 90% of the total fatty acids in the berry seed oils with a much greater n-3/n-6 ratio than most of the commonly consumed vegetable oils. In addition, the seed oils are rich in natural antioxidants. The results from this study may provide a scientific basis for improved utilization of these seed oils for human nutrition and health, especially important for their potential in dietary intake of bioavailable α-linolenic acid and improving the balance between dietary n-3 and n-6 PUFAs. Further animal and pilot human studies could be designed to investigate the nutritional values and health benefits of these berry seed oils in the future.

## Figures and Tables

**Figure 1 foods-10-02530-f001:**
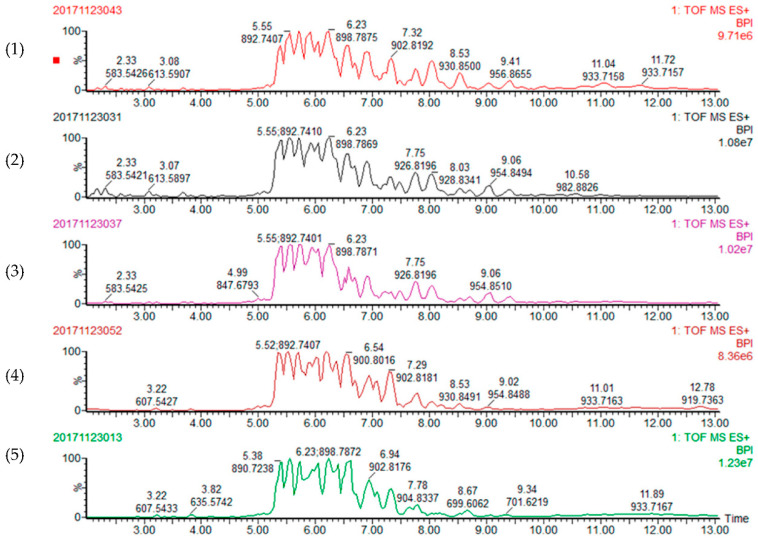
Representative UPC^2^-QTOF MS base peak intensity (BPI) chromatograms of the triacylglycerols from (**1**) blackberry, (**2**) red raspberry, (**3**) black raspberry, (**4**) blueberry and (**5**) cranberry seed oil samples.

**Figure 2 foods-10-02530-f002:**
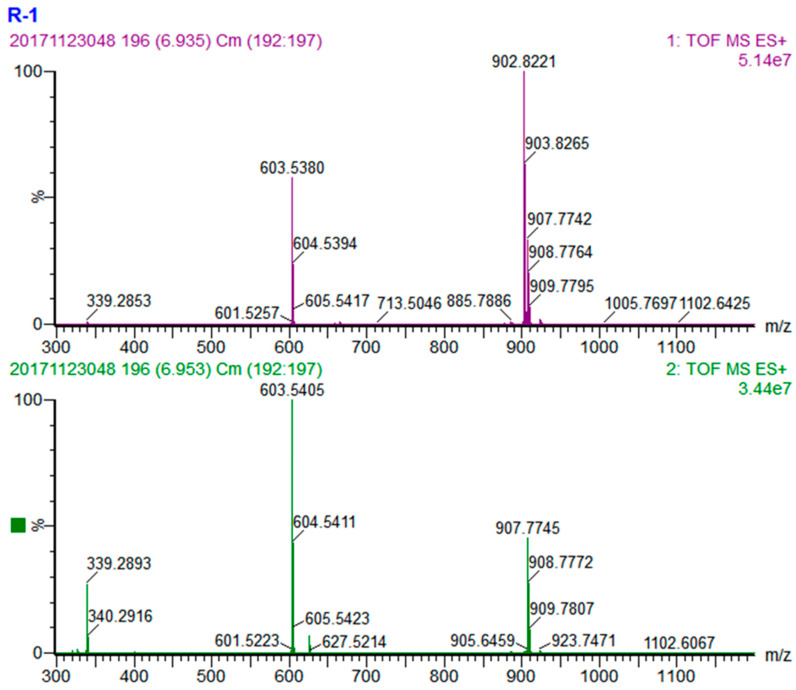
MS spectra of OOO. (**Upper**) MS1 spectrum and (**lower**) MS2 spectrum.

**Figure 3 foods-10-02530-f003:**
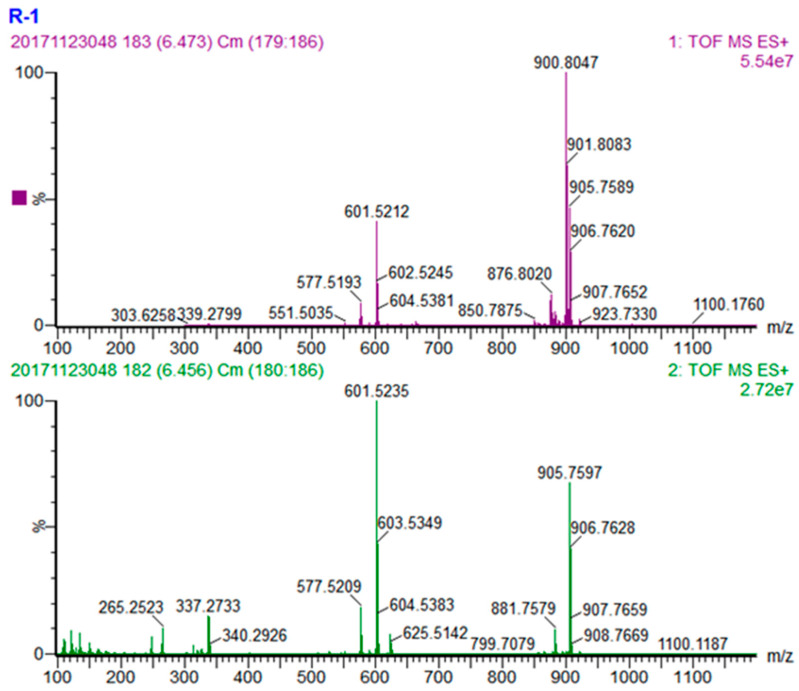
MS spectra of LOO. (**Upper**) MS1 spectrum and (**lower**) MS2 spectrum.

**Figure 4 foods-10-02530-f004:**
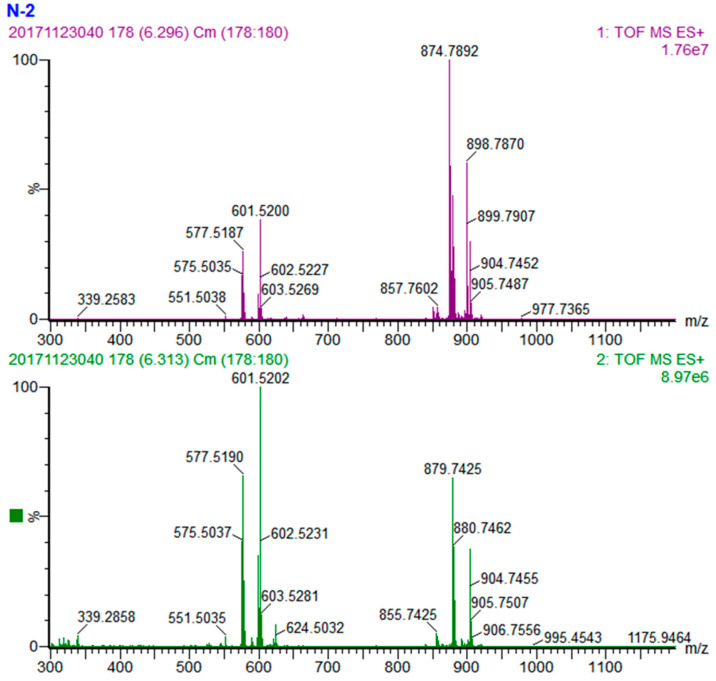
MS spectra of POL. (**Upper**) MS1 spectrum and (**lower**) MS2 spectrum.

**Figure 5 foods-10-02530-f005:**
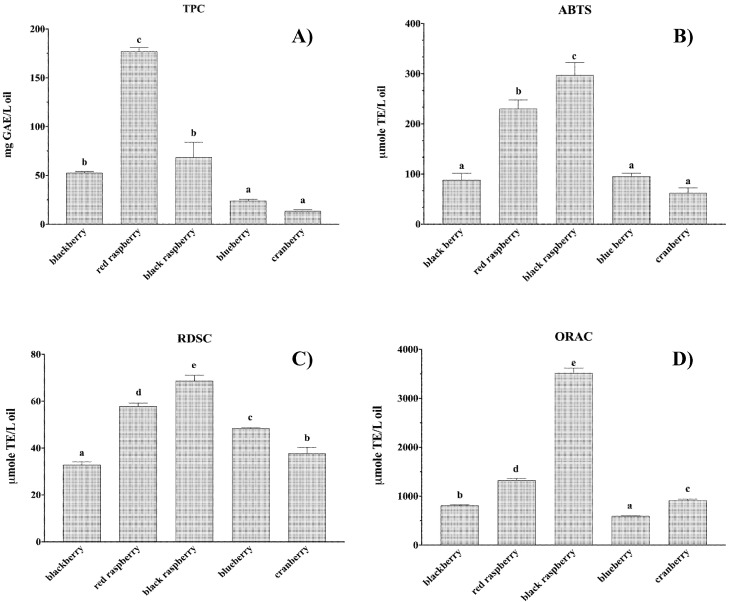
(**A**) Total phenolic content (TPC), (**B**) ABTS radical cation scavenging capacity, (**C**) relative DPPH radical scavenging capacity (RDSC), and (**D**) oxygen radical absorbance capacity (ORAC) of the blackberry, red raspberry, black raspberry, blueberry and cranberry seed oil extracts. Different letters represent significant differences within a column (*p* < 0.05).

**Table 1 foods-10-02530-t001:** Identification and relative concentration of triacylglycerols in five berry seed oils.

RT(min)	Mass ([M + NH_4_]^+^)	Molecular Formula	Possible Structure	UB	TAG Composition (g/100 g TAGs)				
					Blackberry	Red Raspberry	Black Raspberry	Blueberry	Cranberry
4.46	816.7073	C_51_H_90_O_6_	LaLL	4	0.04 b ± 0.00	0.06 c ± 0.01	0.04 b ± 0.00	0.01 a ± 0.00	0.01 a ± 0.00
4.81	840.7066	C_53_H_90_O_6_	MLnLn	6	0.06 a ± 0.01	0.13 b ± 0.02	0.23 c ± 0.00	0.09 a ± 0.00	0.07 a ± 0.00
4.99	842.7233	C_53_H_92_O_6_	MLLn	5	0.20 a ± 0.02	0.36 c ± 0.03	0.44 d ± 0.02	0.27 b ± 0.01	0.20 a ± 0.02
5.17	844.7394	C_53_H_94_O_6_	MLL	4	0.44 c ± 0.03	0.43 c ± 0.01	0.39 b ± 0.01	0.34 b ± 0.02	0.16 a ± 0.01
5.23			MOLn		nd	nd	nd	nd	0.12 a ± 0.01
5.44	846.7556	C_53_H_96_O_6_	MOL	3	0.04 b ± 0.00	0.01 a ± 0.00	0.01 a ± 0.00	0.05 b ± 0.00	0.04 b ± 0.00
6.05	848.7701	C_53_H_98_O_6_	PLP	2	0.09 b ± 0.00	0.04 a ± 0.00	0.03 a ± 0.00	0.04 a ± 0.00	0.03 a ± 0.00
6.40	850.7859	C_53_H_100_O_6_	POP	1	0.07 b ± 0.00	0.03 ab ± 0.00	0.01 a ± 0.00	0.06 b ± 0.00	0.04 ab ± 0.00
4.88	854.7233	C_54_H_92_O_6_	PeLLn	6	0.03 a ± 0.00	0.09 c ± 0.01	0.05 b ± 0.00	0.02 a ± 0.00	0.06b ± 0.00
5.23	868.7403	C_55_H_94_O_6_	PoLnL	6	0.62 a ± 0.02	0.84 b ± 0.05	0.62 a ± 0.03	0.64 a ± 0.02	0.56 a ± 0.04
5.59			PLnLn		0.57 c ± 0.03	0.49 b ± 0.01	0.29 a ± 0.01	1.29 d ± 0.02	1.75 e ± 0.04
5.76	870.7551	C_55_H_96_O_6_	PLLn	5	1.08 b ± 0.01	1.02 b ± 0.09	0.73 a ± 0.01	1.88 c ± 0.02	2.21 d ± 0.07
5.44			PoLL		0.42 c ± 0.01	0.16 b ± 0.00	0.09 a ± 0.00	0.17 b ± 0.01	nd
5.94			PLL		2.45 c ± 0.04	1.24 a ± 0.16	1.19 a ± 0.01	1.49 b ± 0.30	1.22 a ± 0.08
6.08	872.7703	C_55_H_98_O_6_	POLn		nd	0.54 b ± 0.00	0.43 a ± 0.01	1.62 c ± 0.03	2.05 d ± 0.03
6.30	874.7861	C_55_H_100_O_6_	POL	5	1.84 c ± 0.04	1.28 b ± 0.08	1.02 a ± 0.03	2.09 d ± 0.02	1.79 c ± 0.12
7.04	876.8019	C_55_H_102_O_6_	LPS	2	0.43 c ± 0.00	0.17 ab ± 0.01	0.13 a ± 0.00	0.22 b ± 0.01	0.13 a ± 0.00
6.65			POO		0.49 b ± 0.01	0.29 a ± 0.02	0.18 a ± 0.01	1.33 c ± 0.01	1.51 d ± 0.04
7.47	878.8129	C_55_H_104_O_6_	POS	1	0.29 b ± 0.01	nd	nd	0.29 b ± 0.02	0.13 a ± 0.01
5.33	890.7241	C_55_H_96_O_6_	LnLnLn	9	5.02 a ± 0.12	7.36 c ± 0.19	8.85 d ± 0.21	6.72 b ± 0.07	6.35 b ± 0.17
5.52	892.7397	C_57_H_92_O_6_	LnLnL	8	8.51 a ± 0.12	9.04 ab ± 0.74	9.95 b ± 0.49	8.41 a ± 0.30	8.41 a ± 0.55
5.70	894.7553	C_57_H_96_O_6_	LLLn	7	8.66 b ± 0.15	9.67 c ± 0.56	11.67 d ± 0.42	9.71 c ± 0.11	7.87 a ± 0.24
6.44	896.7697	C_57_H_98_O_6_	LnSLn	6	1.04 a ± 0.06	1.45 b ± 0.27	1.43 b ± 0.07	1.08 a ± 0.01	0.90 a ± 0.06
5.91			LLL		8.24 c ± 0.10	6.89 b ± 0.28	9.87 d ± 0.33	4.13 a ± 0.01	nd
6.05			LOLn		4.57 a ± 0.27	6.03 bc ± 0.32	5.56 b ± 0.37	6.33 c ± 0.14	10.65 d ± 0.10
6.19	898.7862	C_57_H_100_O_6_	OLL	5	12.31 c ± 0.24	14.97 d ± 0.42	11.89 c ± 0.49	8.11 b ± 0.21	7.37 a ± 0.16
6.68			LSLn		2.75 ab ± 0.04	3.58 c ± 0.08	3.41 c ± 0.35	2.99 b ± 0.02	2.56 a ± 0.03
6.36			OOLn		nd	nd	nd	4.66 a ± 0.10	6.08 b ± 0.38
6.58	900.8047	C_57_H_102_O_6_	LOO	4	5.40 a ± 0.12	6.79 b ± 0.13	5.41 a ± 0.18	9.54 c ± 0.14	11.64 d ± 0.27
7.07			SOLn		nd	nd	nd	3.39 b ± 0.21	2.91 a ± 0.15
6.9			SLL		5.39 d ± 0.09	5.60 d ± 0.20	4.13 c ± 0.16	3.12 b ± 0.22	2.03 a ± 0.00
7.32	902.8221	C_57_H_104_O_6_	SOL	3	3.90 b ± 0.09	2.51 a ± 0.14	nd	4.26 c ± 0.07	4.00 bc ± 0.14
6.94			OOO		nd	nd	nd	4.35 a ± 0.11	6.54 b ± 0.16
7.32			SLO		nd	nd	1.55 a ± 0.16	nd	nd
8.21	904.8322	C_57_H_106_O_6_	SSL	2	0.81 c ± 0.02	0.23 b ± 0.01	0.15 a ± 0.00	0.17 a ± 0.01	nd
7.78			OOS		1.18 c ± 0.04	0.32 b ± 0.01	0.18 a ± 0.00	2.32 e ± 0.03	1.70 d ± 0.03
8.74	906.8494	C_57_H_108_O_6_	SSO	1	0.26 c ± 0.01	0.05 a ± 0.00	nd	0.12 b ± 0.00	0.07 a ± 0.00
6.79	924.8030	C_59_H_108_O_6_	LLF		0.53 c ± 0.01	0.39 b ± 0.04	0.81 d ± 0.08	0.16 a ± 0.01	0.43 bc ± 0.03
7.00			LnLG		0.61 b ± 0.04	0.69 b ± 0.03	1.30 c ± 0.07	0.38 a ± 0.00	0.61 b ± 0.03
7.47			LnLnA		1.06 c ± 0.02	1.65 d ± 0.13	1.67 d ± 0.10	0.72 b ± 0.01	0.33 a ± 0.03
7.22	926.8173	C_59_H_104_O_6_	LLG	5	2.29 d ± 0.13	1.23 b ± 0.12	1.78 c ± 0.04	0.69 a ± 0.06	1.31 b ± 0.05
7.39			LnOG		nd	nd	nd	0.30 a ± 0.03	0.71 b ± 0.04
7.75			LnLA		2.75 c ± 0.10	3.30 d ± 0.28	3.37 d ± 0.31	1.12 b ± 0.01	0.68 a ± 0.01
8.21	928.8326	C_59_H_106_O_6_	LnOA	4	nd	nd	nd	0.57 b ± 0.03	0.42 a ± 0.03
7.64			LOG		1.59 b ± 0.11	1.02 a ± 0.08	1.04 a ± 0.12	0.97 a ± 0.04	1.59 b ± 0.07
8.03			LLA		5.11 e ± 0.05	3.35 d ± 0.20	3.09 c ± 0.37	0.94 b ± 0.03	0.48 a ± 0.03
8.10	930.848	C_59_H_108_O_6_	OOG	3	nd	nd	nd	nd	0.62 a ± 0.01
8.53			LOA		2.25 d ± 0.04	1.06 c ± 0.10	0.83 b ± 0.03	0.87 b ± 0.06	0.51 a ± 0.01
9.06	932.8641	C_59_H_110_O_6_	OOA	2	0.60 d ± 0.02	0.13 ab ± 0.01	0.09 a ± 0.00	0.24 c ± 0.00	0.18 b ± 0.00
9.63			LSA		0.49 d ± 0.02	0.12 b ± 0.01	0.09 b ± 0.00	0.03 a ± 0.00	0.02 a ± 0.00
8.71	942.8499	C_60_H_108_O_6_	OLHo	4	nd	nd	nd	0.11 b ± 0.00	0.05 a ± 0.00
8.67			LLHn		0.33 a ± 0.01	0.31 a ± 0.03	0.31 a ± 0.01	nd	nd
8.71	952.8345	C_61_H_106_O_6_	LnLnB	6	0.46 c ± 0.01	0.87 d ± 0.08	1.03 e ± 0.09	0.32 b ± 0.02	0.14 a ± 0.00
9.02			LnLB		1.11 c ± 0.05	1.41 d ± 0.17	1.65 d ± 0.23	0.47 b ± 0.02	0.20 a ± 0.01
9.37	956.864	C_61_H_110_O_6_	LLB	4	1.59 e ± 0.02	0.93 c ± 0.12	1.13 d ± 0.13	0.24 b ± 0.01	0.12 a ± 0.00
9.59			LnOB		nd	0.23 b ± 0.03	0.23 b ± 0.00	0.13 a ± 0.00	0.09 a ± 0.00
9.95	958.8807	C_61_H_112_O_6_	LOB	3	0.58 d ± 0.02	0.32 c ± 0.04	0.28 c ± 0.01	0.15 b ± 0.00	0.08 a ± 0.00
10.65	960.8965	C_61_H_114_O_6_	LSB	2	0.16 b ± 0.01	0.05 a ± 0.00	0.04 a ± 0.00	0.04 a ± 0.00	0.04 a ± 0.00
10.16	970.8806	C_62_H_112_O_6_	LLT	4	0.33 c ± 0.01	0.19 b ± 0.02	0.22 b ± 0.01	0.05 a ± 0.00	0.03 a ± 0.00
10.16	980.8647	C_63_H_110_O_6_	LnLnLi	6	0.17 b ± 0.01	0.25 c ± 0.02	0.22 c ± 0.00	0.07 a ± 0.00	0.07 a ± 0.00
10.54	982.8818	C_63_H_112_O_6_	LnLLi	5	0.30 b ± 0.01	0.43 c ± 0.02	0.43 c ± 0.01	0.07 a ± 0.00	0.08 a ± 0.00
11.01	984.8955	C_63_H_114_O_6_	LLLi	4	0.43 c ± 0.01	0.33 b ± 0.02	0.39 bc ± 0.01	0.04 a ± 0.00	0.04 a ± 0.00
12.42	1010.9115	C_65_H_116_O_6_	LnLH	5	0.03 b ± 0.00	0.09 c ± 0.01	0.09 c ± 0.00	0.01 a ± 0.00	0.02 a ± 0.00

TAGs, triacylglycerols; RT, retention time; UB, number of unsaturation; nd, not detectable; P, palmitic acid; M, myristic acid; O, oleic acid; S, stearic acid; L, linoleic acid; Ln, linolenic acid; G, gondoic acid; A, arachidic acid; B, behenic acid. X-X-Y, X-Y-X, X-X-X and X-Y-Z represent the structures of triacylglycerols; for example, SPO stands for the structure of 1/3-stearoyl-2-palmitoyl-1/3-oleoylglycerol. The relative concentration of each triacylglycerol is reported as gram of triacylglycerols/100 g of oil samples. Cranberry, red raspberry, black raspberry, blackberry and blueberry seed oils were analyzed in triplicates and results reported as mean ± standard deviation (SD). Different letters represent significant differences within a column (*p* < 0.05).

**Table 2 foods-10-02530-t002:** The fatty acid compositions in the five berry seed oils.

Fatty Acid	C:D	Fatty Acid Composition (g/100 g Total FAs)				
		Blackberry	Red Raspberry	Black Raspberry	Blueberry	Cranberry
palmitic acid	16:0	5.63 c ± 0.03	3.70 b ± 0.08	2.40 a ± 0.04	7.47 d ± 0.10	7.68 d ± 0.02
stearic acid	18:0	2.81 d ± 0.02	1.39 b ± 0.06	0.84 a ± 0.01	2.38 c ± 0.05	1.56 b ± 0.03
oleic acid	18:1	1.34 a ± 0.06	1.36 a ± 0.14	1.17 a ± 0.04	1.15 a ± 0.09	1.13 a ± 0.04
linoleic acid	18:2	59.12 f ± 0.26	52.37 e ± 0.30	51.03 d ± 0.28	40.20 c ± 0.12	38.40 b ± 0.16
linolenic acid	18:3	31.10 a ± 0.28	41.18 b ± 0.51	44.56 c ± 0.33	48.80 d ± 0.34	51.23 e ± 0.16
SFA		8.44	5.09	3.24	9.85	9.24
UFA		91.56	94.91	96.76	90.15	90.76
MUFA		1.34	1.36	1.17	1.15	1.13
PUFA		90.22	93.55	95.59	89.00	89.63

FAs, fatty acids; C:D, carbon number: double bounds number; nd, not detectable; SFA, saturated fatty acids; UFA, total unsaturated fatty acids; MUFA, monounsaturated fatty acids; PUFA, polyunsaturated fatty acids. The relative concentration of each fatty acid is reported as gram of fatty acids/100 g of total fatty acids. All the seed oils were analyzed in triplicates and results reported as mean ± standard deviation (SD). Different letters represent significant differences within a column (*p* < 0.05).

## Data Availability

Not applicable.

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
