# Peer review of "Triacylglycerol and Fatty Acid Compositions of Blackberry, Red Raspberry, Black Raspberry, Blueberry and Cranberry Seed Oils by Ultra-Performance Convergence Chromatography-Quadrupole Time-of-Flight Mass Spectrometry"

_foods, 2021, doi:10.3390/foods10112530_

Round 1
Reviewer 1 Report
Dear Authors,
in my opinion article entitled "Triacylglycerol and fatty acid compositions of blackberry, red raspberry, black raspberry, blueberry and cranberry seed oils by ultra-performance convergence chromatography-quadrupole time-of-flight mass spectrometry" can not be accepted for publication in present form. All described oils are well known ant their composition has been described many times. From a nutritional point of view it doesn't matter what the composition of the triacylglycerols is. Fatty acids released under the influence of lipases in the digestive process are important, and their composition in the described oils is already and well known. If the manuscript was focused on a relatively new method and its application in fat analysis, but this requires practically rewriting of the article. I suggest rebuilding the manuscript and resending it for review as a independent text.
Author Response
Appreciate the comment. We do agree that there are previous studies about these berry seed oils. But as we mentioned, all these previous studies were focused on free fatty acid compositions of seed oils. None of them investigated the triacylglycerol compositions. And the structure of triacylglycerol, especially the sn-position of fatty acids are important. Since similar fatty acid composition might play significant different nutritional values while in different sn-position, just like the references we cites about the sn-position of palmitic acid in infant formula. We have revised this section in the ‘Introduction’ to make it more clear: “But all these previous studies about berry seed oil compositions were focused on their fatty acid composition. Actually, none of those previous studies investigated the TAG composi-tions with sn-location information for the n-3 and other fatty acids. It is known that the sn-position of fatty acids in the TAGs plays a very important role in determining their ab-sorption and bioavailability, and consequently their nutraceutical values [22–25]. For example, the sn-position of palmitic acid in triacylglycerol of milk could significantly change the lipids and calcium absorption, thus affect fecal conditions of infant [23-24]. Therefore, understanding the sn-positions of the fatty acids is important for their func-tionality and behavior of the TAG in food systems.” (Line 65-73).
Reviewer 2 Report
Many formal and typing errors – see below for a few examples:
Line 36-37: except for their sweet and sour tastes …
Line 52: van Hoed et al. investigated ….
Line 56: .. they were two research groups included in one work ?
Line 59: What does it mean ? [16,2-,21].
Line 75: … reagents …
Line 83: … containing 37 FAMEs
Line 221-222: MUFA usually means monounsaturated fatty acids ?!
Line 249-250: … minor amounts of fatty acids …….. might not be detected by GC-MS due to the detection limits of the two methods…….. These detection limits should be mentioned.
Line 277-278: These results indicated that different berry seed oils might contain different types of antioxidants. Are there some publications supporting this statement ? Should be discussed here.
Author Response
General comment: Many formal and typing errors – see below for a few examples:
Answer: Thanks for all these comments, and we have careful checked, revised all the errors based on reviewer’s comment.
Comment 1: Line 36-37: except for their sweet and sour tastes …
Answer: Done (Line 38 and 39).
Comment 2: Line 52: van Hoed et al. investigated ….
Answer: Done (Line 58).
Comment 3: Line 56: .. they were two research groups included in one work ?
Answer: the whole sentence has been revised to “In 2014, Radočaj et al. reported the PUFA contents of …” (Line 62).
Comment4: Line 59: What does it mean ? [16,2-,21].
Answer: It has been revised to [16-21] (Line 65).
Comment5: Line 75: … reagents …
Answer: Done (Line 85).
Comment 6: Line 83: … containing 37 FAMEs
Answer: Done (Line 93).
Comment 7: Line 221-222: MUFA usually means monounsaturated fatty acids ?!
Answer: Yes, MUFA is the abbreviation of monounsaturated fatty acid. No change was made.
Comment 8: Line 249-250: … minor amounts of fatty acids …….. might not be detected by GC-MS due to the detection limits of the two methods…….. These detection limits should be mentioned.
Answer: Appreciate the comment. The comparison between these two methods has been added. “Since the limit of detection of fatty acids in GC method are generally at microgram per milliliter, which are much higher than that in UPC2-MS.” (Line 288-290).
Comment 9: Line 277-278: These results indicated that different berry seed oils might contain different types of antioxidants. Are there some publications supporting this statement ? Should be discussed here.
Answer: The whole sentence has been revised to “These results indicated that different berry seed oils might contain different antioxidant capabil-ities and differed in their interacitons with individual radical systems.” to made it clear (Line 317-318).
Reviewer 3 Report
The purpose of this study was to look at the composition of the TAG profile in a number of different berry seed oils. To analyze the TAG profile the researchers used UPC-QTOF MS. The researchers also looked at GAE, total phenolic content and free radical scavenging activity.
I would recommend you have someone reread the paper for grammatical and flow errors.
Line 44 - 45: I would recommending removing their to help improve the flow of the sentence.
Line 49: I recommending adding Berry seeds are known "to be" rich...
Line 157 - 158: I would reorganize the sentence to read with the same three...
Author Response
Comment 1: The purpose of this study was to look at the composition of the TAG profile in a number of different berry seed oils. To analyze the TAG profile the researchers used UPC-QTOF MS. The researchers also looked at GAE, total phenolic content and free radical scavenging activity.
I would recommend you have someone reread the paper for grammatical and flow errors.
Answer: Appreciate the comment. We have carefully check the whole manuscript, and revised grammatical and flow errors (Line 38-39, 58, 62, 65, 85, 93, etc).
Comment 2: Line 44 - 45: I would recommending removing their to help improve the flow of the sentence.
Answer: The whole sentence has been revised to “Except for the sweet and sour tastes, berry fruits are well welcomed for particular nutraceutical values in delaying the aging process and reducing the risk of several human chronic diseases” (Line 39-41).
Comment 3: Line 49: I recommending adding Berry seeds are known "to be" rich...
Answer: Done (Line 55).
Comment 4: Line 157 - 158: I would reorganize the sentence to read with the same three...
Answer: Appreciate the comment. The whole sentence has been revised to “The peak at the retention time 6.94 min is a representative TAG with similar fatty acids in all the three sn-positions” (Line 191-192).
Round 2
Reviewer 1 Report
Due to the amendments made, I believe that the article in its current form may be published in "Foods".
Author Response
Appreciate the comment
Reviewer 2 Report
Comments to author
Line 47-48: wines, beverages, jams and smoothies ….
Line 56-57: cranberry, blueberry and strawberry seed oils ….
Line 59: ….of α-linolenic acid …
Line 62-63: … also detected high n-3 and total PUFA content in berry seed oils ..
Line 88: … was food grade …
Line 248: … over 90% of unsaturated fatty acids on a per weight basis, and less than 10% of saturated fatty …
Line 255-256: and created about 50% of the total fatty acids in the ……, followed by linoleic acid at a level ….
Line 259: … cannot be detected by GC-MS …
Author Response
Comment 1: Line 47-48: wines, beverages, jams and smoothies …
Answer: Done (Line 47).
Comment 2: Line 56-57: cranberry, blueberry and strawberry seed oils ….
Answer: Done (Line 55).
Comment 3: Line 59: ….of α-linolenic acid …
Answer: Done (Line 58).
Comment4: Line 62-63: … also detected high n-3 and total PUFA content in berry seed oils ..
Answer: Done (Line 61).
Comment5: Line 88: … was food grade …
Answer: Done (Line 86).
Comment 6: Line 248: … over 90% of unsaturated fatty acids on a per weight basis, and less than 10% of saturated fatty …
Answer: Done (Line 244).
Comment 7: Line 255-256: and created about 50% of the total fatty acids in the ……, followed by linoleic acid at a level ….
Answer: Done (Line 250 and 251).
Comment 8: Line 259: … cannot be detected by GC-MS …
Answer: Done (Line 254).